# The Methods of Three-Dimensional Modeling of the Hydrogenerator Thrust Bearing

**Oleksii Tretiak [1,\*], Dmitriy Kritskiy [2,\*] , Igor Kobzar [3,\*], Mariia Arefieva [4,\*] and Viacheslav Nazarenko [1,\*]**

1   Faculty of Aircraft Engineering, Department of Aerohydrodynamics, National Aerospace University Kharkiv Aviation Institute, 61070 Kharkov, Ukraine
2   Department of Information Technology Design, National Aerospace University Kharkiv Aviation Institute, 61070 Kharkov, Ukraine
3   Special Design Office for Turbogenerators and Hydrogenerators, Join Stock Company "Ukrainian Energy Machines", 61037 Kharkov, Ukraine
4   Kharkiv Lyceum "IT Step School Kharkiv", 61010 Kharkov, Ukraine
\*   Correspondence: alex3tretjak@ukr.net (O.T.); d.krickiy@khai.edu (D.K.); ivkobzar@ukr.net (I.K.); marii.arefieva@gmail.com (M.A.); my_registrator@ukr.net (V.N.)

**Abstract:** In the presented scientific work, the basic design versions of the thrust bearings of Hydrogenerators are considered. The main causes of emergencies in the thrust bearing unit of a high-power Hydrogenerator are considered. The main requirements for the operation of thrust bearings are submitted. Cause-and-effect relationships of emerging and development of defects are established. Existing methods for calculating the stressed state of a thrust bearing in the classical formulation for a stationary mode of operation are considered. The main features of the operation of the thrust bearing unit are investigated in relation to the features of the sliding bearings. The calculation of the elastic chambers of the hydraulic thrust bearing in a three-dimensional formulation is carried out, taking into account the physical properties of the oil, the material of the chambers and distribution of the acting loads. It is shown that the applied designs of Join Stock Company "Ukrainian Energy Machines" can be used in high-power Hydrogenerators.

**Keywords:** the thrust bearing; fatigue calculation; three-dimensional modeling; fatigue curve

## 1. Introduction

Hydrogenerators like other electrical machines consist of active and constructional parts. Active parts directly involved in to the process of energy conversion including the rotor and stator magnetic cores circuits with windings on them. All other elements are called constructive ones. The construction of the vertical Hydrogenerator is presented in the scientific work of A. I. Abramov [1].

A characteristic feature of the construction of vertical Hydrogenerator is the presence of the support bearing, called the thrust bearing. It reliably operates under the influence of large forces created by the masses of the rotor of Hydrogenerator and the turbine rotor, as well as the vertical component of the water pressure acting on the turbine runner. These forces are transferred through the thrust bearing bush fixed on the shaft and the thrust bearing on the load bearing or supporting spider and then through the stator body to the foundation.

As a rule, the main units of large electrical machines operate with loads, the mechanical stresses of which are significantly removed from the maximum permissible. Therefore, the probability of fatigue failure appears very rarely. However, when they appear, it leads to significant structural damage or to the replacement of existing support structures (the refurbishment time takes an extended period). A significant contribution in to the solution of the problem of determining the fatigue stresses for the main generating equipment was made by Chernousenko O. Yu. [2].

The thrust bearings operate in the oil friction mode. They are characterized by exceptional durability and low friction coefficients.

The calculation of thrust bearings on rigid supports is submitted in the paper [3,4]. A significant advantage of the work is the consideration of the oil wedge. However, given thrust bearings are not used in the design of high-power hydrogenerating units, as they have limitations on the ultimate load on the support planes.

In the paper [5], the types of failures of hydrogenerating units are submitted in detail, however, its usage is limited by the lack of a three-dimensional statement of the problem under consideration.

In the paper [6], the causes of damage due to misalignment of the shafts of hydrogenerating units are indicated, the presented work has limitations in application, due to the fact that for long shafts of hydrogenerating units, the runout is monitored by the control system, and work with unacceptable vibrations is prohibited. In general, this algorithm, specified in the paper [7], can be adapted to large machines, to study the operation after 30 years of operation (over-design mode of operation).

In the paper [8], the vibrations of hydrogenerating units were studied; it should be noted the high reliability of the obtained information. An undoubted advantage is the study of vibrations of a real design structure.

When designing new hydrogenerating units, it is necessary to take into consideration the experience of the paper [9] of Michell Bearings for 65 years of its existence, which combines practical innovations with modern design technologies.

At the same time, when calculating the oil wedge in the bearing surface, it is necessary to consider and pay a special attention to the paper [10], as it allows you to answer the issues of the correct choice of gaskets a prototype set of PTFE pads.

In the paper [11] schematic and power solutions for the thrust bearing of an operating hydrogenerating unit are submitted. In this case, the obtained dependencies were correlated by the author with the experiment. And the paper [12] regulates in detail the assembly process and restrictions on the operation of hydro-electric units. In the paper [13], a research facility was developed and manufactured to study the design and opereational characteristics of industrial hydrodynamic thrust bearings, which allows verifying the results of previously presented scientific studies.

In the paper [14], a team of authors performed a three-dimensional analysis of the stress-strain state of a thrust bearing on rigid supports. At the same time, the presence of a thermoelastic component shall be considered as a significant advantage of the work. However, due to the fact that the thrust bearings on hydraulic supports include vessels under high pressure, this algorithm is not applicable for hydrogenerating units with a power over than 300 MW. The paper [15] deserves a special attention in terms of modeling the flow of the oil layer, while paper [16] can be supplemented with the 3-rd (the third) boundary conditions obtained from the results of the paper [15].

The invention of inclined sliding bearings A. G. M. Michell [16] in Australia and independently, A. Kingsbury [17] in the USA let use existing designs during decades.

However, during the last time there was a sharp jump in technical and design capabilities, which allowed creating qualitatively new products.

In Figure 1 existing designs Waukesha Bearings and Itaipu Power Plant are submitted, where 1 is the thrust bush (skirt), 2 is the shaft of Hydroaggregate, 3 is cooler, 4 is the thrust segment.

One of the options for ensuring the stability of the thrust bearing operation is the introduction of an additional layer into the friction zone (see Figure 2).

However, given design does not solve completely all the technical problems as bronze is electrically conducting material.

The elastic chambers with corrugation are used in the original design of the thrust bearing unit of JSC "Ukrainian Energy Machines". Such design is described in scientific work [18]. Given design solution let provide reliable operation of the thrust bearing unit with loads of 300 t. Due to the fact that Hydroaggregates should operate not less than

40 years and operation conditions demand high maneuverability of the electrical machines then it is necessary to carry out a fatigue analysis of the thrust bearing unit operation for the modes where the service life should overcome the claimed one.

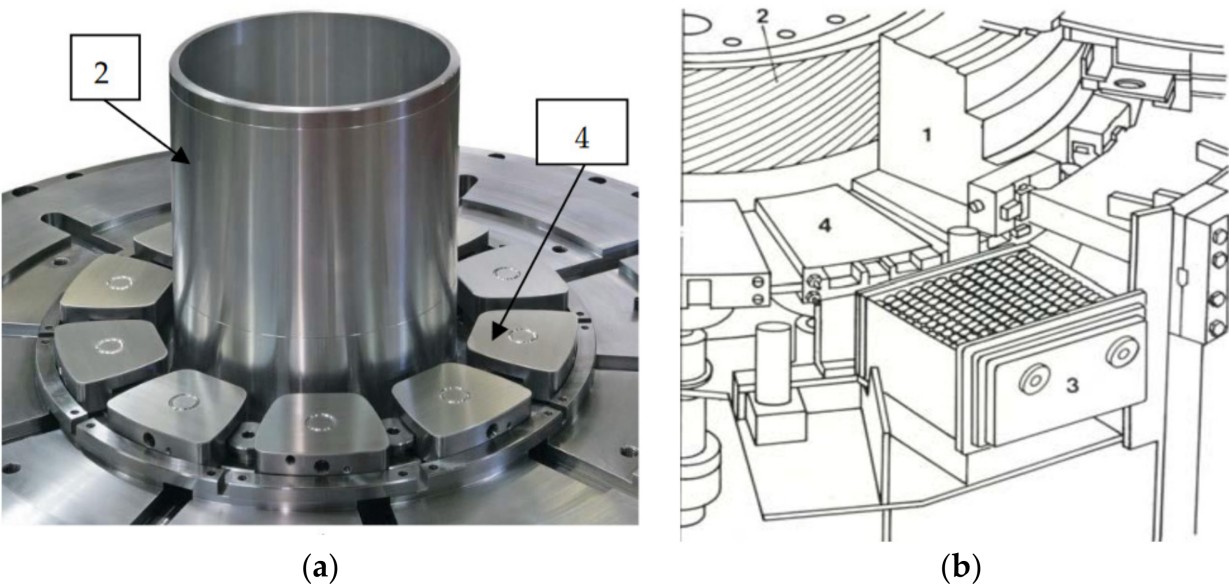

(**a**)  (**b**)

**Figure 1.** Tilting pad thrust bearing: (**a**) courtesy of Waukesha Bearings; (**b**) courtesy of Itaipu Power Plant.

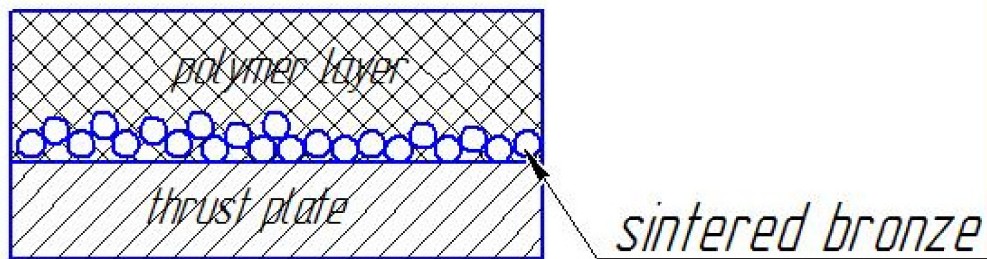

**Figure 2.** Location of friction with the bronze-plated.

### 1.1. The Main Causes of Damage of the Thrust Bearing Chambers

The main cause of damage of the thrust bearing chambers of the vertical Hydrogenerator is depressurization of chambers at that the thrust bearing from hydraulic with automatic load balancing between the segments turns into the thrust bearing on rigid supports. However, as a rule, there is a significant misalignment of the load on the segments.

In Figure 3 the Diagram of development of the above-mentioned defect of the thrust bearing on the hydraulic support is shown. The causes of the defect (they are shown at the 1-st and 2-nd time levels) are increased pulsation in the thrust bearing or manufacturing defects. In the time levels 3–7 the further development of the emergency situation is shown.

When cracks appear in the elastic chamber its replacement is necessary followed by filling of whole system with oil. Features of the study of emergency situations are submitted in scientific work [19].

### 1.2. Main Requirements to the Thrust Bearing Unit of Hydroaggregate Rated 330 MW

At operation of Hydrogenerator-Motor rated 330 MW in the generator mode during the start the axial load on to the thrust bearing from the masses of rotating parts of the pump-turbine and water reaction can achieve of 3000 kN and total axial load from water pressure and the rotating parts of the Hydroaggregate at rated mode shall not exceed 23,000 kN.

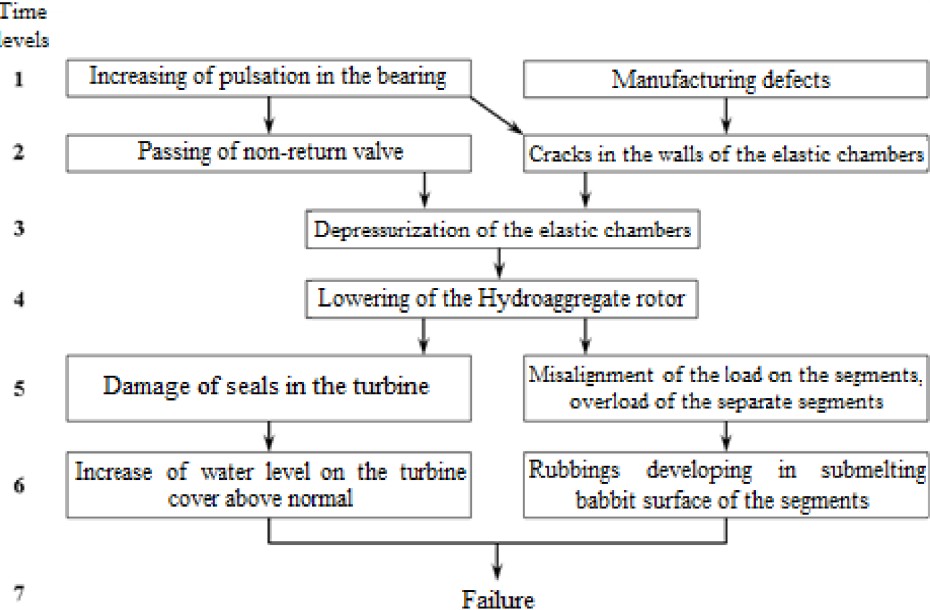

**Figure 3.** The Diagram of Defects Development of the Thrust Bearing Chambers.

At that the requirements to safety and health protection are strictly regulated by the relevant documents. Hydrogenerator for safety of the design shall comply with GOST 12.1.004-91, GOST 12.1.010-76, GOST 12.2.007.0-75 and GOST 12.2.007.1-75. According to the method of protection of a person from electric current shock, Hydrogenerator has class 01 in accordance with GOST 12.2.007.0-75, and its fire safety corresponds to GOST 12.1.004-91. The degree of protection with shell of the Hydrogenerator-Motor shall be IP00 according to Publication IEC 34-5.

In addition, the vertical vibration of the thrust bearing base (double amplitude) shall not exceed of 0.15 mm, and the temperature of the surface of the Hydrogenerator-Motor during inspections and repairs is limited to 45 °C.

At that, the thrust bearing design shall not allow overheating, vibration, oil spraying and getting into the ventilation duct at all operation modes, including start-ups and stops, as well as operation at maximum runaway rotation speed.

Designing of the bearings with a two-row arrangement of the segments, that allowed to maintain the level of the specific load without increasing the size of the segments. In the two-row design of the thrust bearing (two rows of chambers) the load through the hub and disc is transferred to segments located in two circular rows. The segments are laid on the supporting plates, which are supported by the spherical surfaces of the supporting bolts screwed into the balancing beam. The balancer is on a cylindrical support. Segments are connected in pairs by means of the balancing beam. The load between the external and internal segments in the pair is automatically distributed according to the law of the double-sided lever (the first-order lever).

Its disk provides electrical isolation from the housing to prevent leakage of the bearing currents. Insulation monitoring is performed on a stopped Hydrogenerator-motor without disassembling of the bath. The segments of the thrust bearing are made with elastic metal-plastic straps (with fluoroplastic coating).

At that the specific pressure on to the thrust bearing segments are limited with 6 MPa. The design provides uniform distribution of the load on to the segments with tolerance of ±10%.

The thrust bearing is arranged in the oil bath protected from water ingress and is self-lubricating. For its lubrication oil of Grade Tp-30 is used in accordance with GOST 9972-74 (or its analogue). It is cooled with water circulating through the oil coolers arranged in the oil bath to temperatures corresponding to the established norms.

The thrust bearing ensures reliable operation in both directions of rotation of the Hydroaggregate as well as the start of the Hydrogenerator after its long term stop without usage of special measures.

## 2. Determining of Maximum Admissible Operation Time of the Thrust Bearing Unit

The classical approach to calculating of the fatigue state of the thrust bearing unit reduces to taking into account only uniform distribution of the load (tension-compression) over the entire area of the segments, the diagram of which is shown in Figure 4.

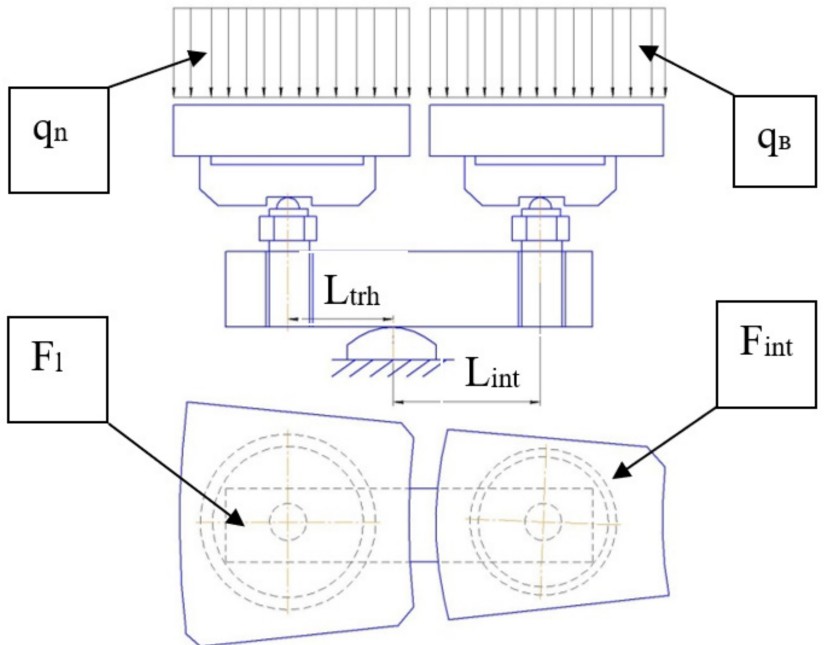

**Figure 4.** Loading Diagram of the double-row thrust bearing segments.

The condition for the equality of the moments of forces acting on the segments shall be written in the following form:

$$P_{ext}l_{ext} = P_{int}l_{int} \text{ or } q_{ext}F_{ext}l_{ext} = q_{int}F_{int}l_{int}, \tag{1}$$

where $P_{ext}$ and $P_{int}$—are the loads on to the external and internal segments; $q_{ext}$ and $q_{int}$—are specific loads on to the working surface of the external and internal segments; $F_{ext}$ and $F_{int}$—are the areas of the working surface of the external and internal segments; $l_{ext}$ and $l_{int}$—the lengths of the arms of the lever of the external and internal segments.

In the Commonwealth of Independent States states more than 70 double-row thrust bearings were manufactured and installed on the aggregates of various hydroelectric power plants. They are designed for the total load from 20 up to 35 MN and specific load from 3.5 up to 4.2 MPa. The average circumferential speed in the internal row of segments comprises of 8.0–24.0 m/s, and on the external one comprises of 10.8–31.0 m/s.

When carrying out the analytical calculation, the rigidity of the entire design is searched for by adding of the chamber rigidity due to the compressibility of oil and the rigidity of the empty chamber:

$$C = 0.85A + B, \tag{2}$$

where *A*—the rigidity of the chamber due to the compressibility of oil; *B*—the rigidity of the empty chamber.

The submitted approach can be correct in the first approximation and used at the stage of preliminary outline design.

However, for the working project the calculation diagram shall be radically revised. The operation peculiarities of the chambers of the near and the far rows are different therefore it is necessary to take into account the entire three-dimensional figure of the loads effect on the thrust bearing that is possible to carry out only at solid state modeling of the entire design in a three-dimensional formulation.

Let us consider the three-dimensional calculation design model of the thrust bearing chamber, made by the finite element method (FEM) (SolidWorks Simulation). The design diagram and the limiting conditions of the thrust bearing unit loading under the action of the total axial load from water pressure and rotating parts of the pump-turbine unit, as well as the mass of the rotating parts of the pump-turbine and the water reaction are submitted in Figure 5.

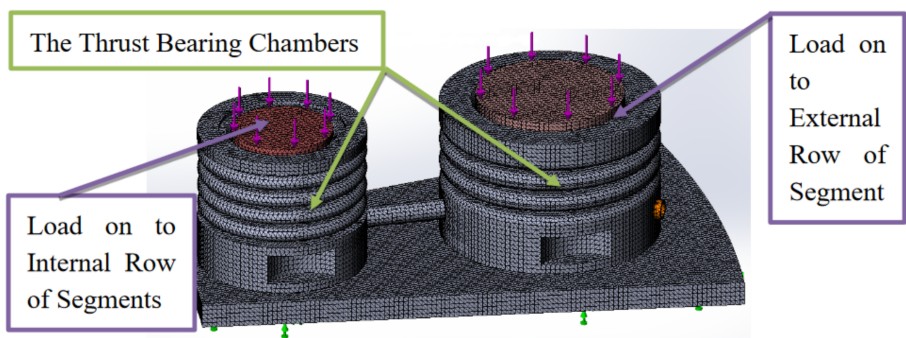

**Figure 5.** Calculation Diagram and limiting conditions of the thrust bearing unit loading.

As it can be seen from Figure 6 the design vessel (the thrust bearing chamber) is closed. It means that the physical properties of oil shall be the same everywhere and, consequently, pressure in all points of the chamber shall be constant ($P = const$).

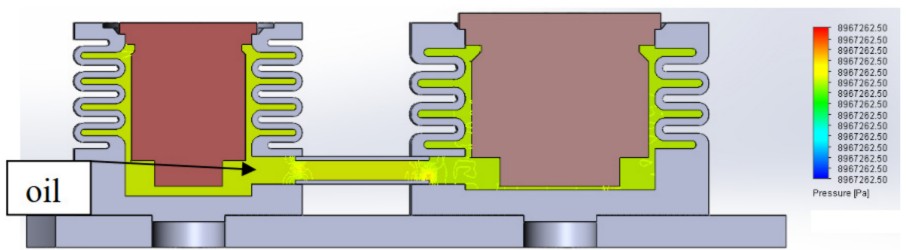

**Figure 6.** Diagrams of pressure inside the thrust bearing chambers.

The successive loading of the thrust bearing was considered at small increments of the load that allowed observing insignificant change in the oil elasticity modulus for each iteration. At that the rigidity of the thrust bearing changes significantly at all stages of operation.

The diagrams of pressure inside the thrust bearing chambers are submitted in Figure 6. The calculations are carried out with the help of the package SolidWorks Flow Simulation. As it can be seen from Figure 6 the values of pressures are practically constant.

When carrying out the fatigue calculation, the following factors shall be taken into account:

- technological factor;
- geometrical factor;
- the surface roughness factor (taking into account additional influence of roughness to local stresses and, consequently, on the fatigue strength of the component);
- the influence factor of surface hardening (taking into account influence (residual stress, hardness) of the changed state of the surface to fatigue strength of the corresponding technological procedure)

Thermally treated steel 35 GOST 1050-2013 shall be considered as the basic material for the chambers of the thrust bearing unit. As its foreign analogues the brands submitted in scientific work [20] can be considered. Check for compliance of all the mechanical properties of the material of the thrust bearing chamber shall be in accordance with GOST 8479-70 for Group V. The fatigue curve equation under rigid loading was performed in double logarithmic coordinates according to GOST 25.507-85.

The fatigue calculation algorithm assumes the choice of the maximum effective load from hydraulic shock and the masses of the rotating parts of the hydrogenerating unit. In this case, a load is set that varies in time depending on the of rotational speed of the hydrogenerating unit. In this case, the full cycle is considered to be full-loading followed by unloading. The maximum stress at which fatigue failure occurs after a given number of load cycles. The permissible number of load cycles is determined based on the regulatory documents for hydrogenerating units (the maximum number of starts). To calculate the compliance of the supports, a correlation of oil elasticity is put depending on the load.

As per the results of the research it can be seen (see Figure 7) that run out (wearing out) of the design should occur much later than the designed operation life of the design.

Percentage of damage.

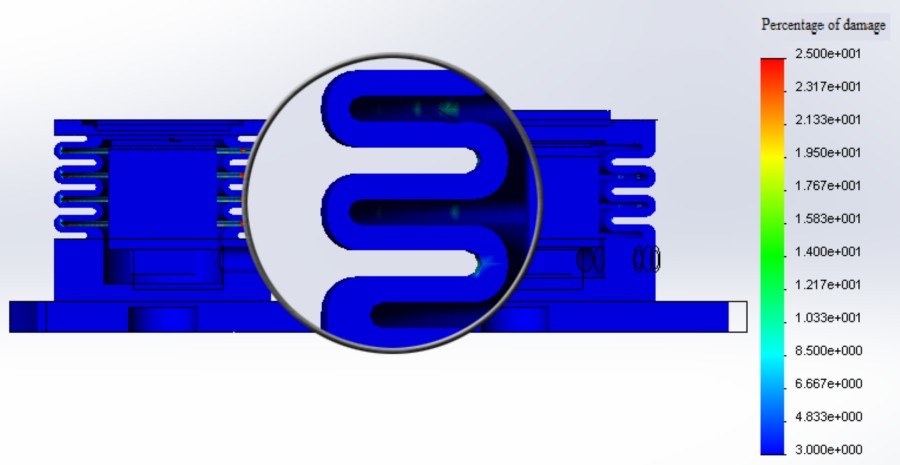

**Figure 7.** Percentage of damage of the thrust bearing chamber.

### 3. Discussion

All over the world, there are 2 types of the thrust bearings: on rigid and hydraulic supports. All existing algorithms are designed for rigid supports, because they are most often used in Hydrogenerators with medium power. For European schools of mechanical calculation, the main trend was a decrease in weight and overall dimensions indices due to an increase in mechanical stresses approaching the allowable ones. This algorithm was normally transferred to body parts and individual parts of the rotor group. However, due to the fact that hydraulic thrust bearings are used on hydro generators of maximum power for these types of bearings, this algorithm has not been fully implemented. Creation in Ukraine of Hydrogenerators with power over 300 MW for the first time made it possible to switch to accurate calculations by the FEM method of thrust bearings, taking into account the properties of the liquid inside the support. In our opinion, Chinese concerns will continue to use similar algorithms for the transition to high power Hydroelectric Units.

### 4. Conclusions

In submitted work, the stressed state of the thrust bearing unit of Hydrogenerator of the limiting power is considered. The main disadvantages of the existing designs are indicated, as well as the most dangerous technical damage. To ensure reliable operation of the Hydroaggregate, a methodology for performing fatigue calculation is proposed, which allows to take into account technological, operational and constructive factors, by

FEM modeling of the design. It is shown that the design under consideration satisfies the requirements of GOST 14965-80, and the required service life for 40 years is reliably ensured.

**Author Contributions:** Writing—Review and editing O.T., D.K., I.K., M.A., V.N. All authors have read and agreed to the published version of the manuscript.

**Funding:** This research received no external funding.

**Institutional Review Board Statement:** Not applicable.

**Data Availability Statement:** Not applicable.

**Conflicts of Interest:** The authors declare no conflict of interest.

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
