# Peer review of "The Methods of Three-Dimensional Modeling of the Hydrogenerator Thrust Bearing"

_computation, doi:10.3390/computation10090152_

Round 1

Reviewer 1 Report

Dear Authors

In the manuscript received for review, the suitability of the JSC "Ukrainian Energy Machines" thrust slide bearing (tilting pad bearing) for use in high-power hydrogenerators was investigated. It was limited to the stationary mode of operation (according to Abstract), although at the same time the Authors mention that a heterogeneous loads were assumed when calculating elastic chambers (in 3D).

Remarks:

1) need to explain what the Authors mean by "non-uniformity of the acting loads" (in Abstract)?

2) the methodology of calculating the critical bearing elements used by the Authors should be presented in Introduction against the background of papers of other authors (sample papers in the attachment),

3) Commonwealth of Independent States should be written instead of CIS, because then an error would be avoided (using states twice) - line 157

4) the method of determining fatigue life should be slightly more detailed - what is the main cause of surface fatigue (under constant load) and which elements it concerns and to what extent (pads-thrust segments or chambers). Maybe the reason is also the rotation of the shaft, maybe its fluctuations, assembly errors, maybe inadequate lubrication (oil pressure or viscosity) or maybe only the indicated factors (line 196)

5) the fatigue strength diagram of "powders based on nickel (line 250) cannot be used to determine the fatigue life of an element made of steel GOST 1050-213 (line 204).

I am taking the liberty of publishing a few items of literature that will allow the Authors to present the contribution of other authors to the development of design calculations of tilted pad bearings and make it easier for novice researchers and designers of such bearings to get acquainted with this difficult issue. I have also included sample papers on the calculation of the hydrodynamic lubrication of such bearings. These are very important calculations in non-stationary work conditions, because then the fastest accumulation of fatigue, thermal and/or wear damage occurs. Of course, these are not all the papers available on the subject, for example on the Internet. 

1) S. M. DeCamillo, A. Dadouche, M. Fillon: Thrust Bearings in Power Generation. In: Wang, Q.J., Chung, YW. (eds) Encyclopedia of Tribology. Springer, Boston, MA. https://doi.org/10.1007/978-0-387-92897-5_57

2) Xin Liu, Yongyao Luo, Zhengwei Wang :A review on fatigue damage mechanism in hydro turbines. Renewable and Sustainable Energy Reviews 54 (2016) 1-14

3) H.Iliev: Failure analysis of hydro-generator thrust bearing. Wear Volumes 225–229, Part 2, April 1999, Pages 913-917

4) Luka Selak, Peter Butala, Alojzij Sluga: Condition monitoring and fault diagnostics for hydropower plants. Computers in Industry 65 (2014) 924-936

5) Wanjun Xu and Jiangang Yang: The Air Lubrication Behavior of a Kingsbury Thrust Bearing Demonstration. Hindawi International Journal of Rotating Machinery Volume 2021, Article ID 6690479, 10 pages https://doi.org/10.1155/2021/6690479

6) R. Nowicki, N. Morozov, et.al.: Thrust bearing monitoring of vertical hydro-turbine-generators. June 2017 Conference: 1st World Congress on Condition Monitoring, London, 13-16 JUNE 2017, Paper #174, pp 15

7) J.E.L.Simmons: Paper II(iii) Michell and the development of tilting pad bearings. Tribology Series Volume 11, 1987, Pages 49-56

8) J. E. L. SimmonsR. T. KnoxW. O. Moss: The development of PTFE (polytetrafluoroethylene)-faced hydrodynamic thrust bearings for hydrogenerator application in the United Kingdom. The Journal of Engineering Tribology Procedings First Published May 1, 1998

https://doi.org/10.1243/1350650981542155

9) Shifu Yang et. al: A comparative experimental study on large size center and bi-directional offset spring-bed thrust bearing. Proc IMechE Part J: J Engineering Tribology 2020, Vol. 234(1) 134–144

10) A GENERAL GUIDE TO THE PRINCIPLES, OPERATION AND TROUBLESHOOTING OF HYDRODYNAMIC BEARINGS. Kingsbury, Inc.

11) L. Dabrowski, M. Wasilczuk: A Method of Friction Torque Measurement for a Hydrodynamic Thrust Bearing. Joutnal of Tribology October 1995, vol. 117, pp. 674-678

12) M. Filon: Thermal and Deformation Effects on Tilting-Pad Thrust and Journal Bearing Performance. IBERTRIB 2005 – III CONGRESSO IBÉRICO DE TRIBOLOGIA.

13) M. FILLON1, M. WODTKE, M. WASILCZUK: Effect of presence of lifting pocket on the THD performance of a large tilting-pad thrust bearing. Friction DOI 10.1007/s40544-015-0092-4

14) C. I. Papadopoulos L. Kaiktsis, M. Fillon: Computational Fluid Dynamics Thermohydrodynamic Analysis of Three-Dimensional Sector-Pad Thrust Bearings With Rectangular Dimples. Journal of Tribology January 2014, vol. 136/011702-1.

Yours sincerely, Reviewer

Author Response

Dear Reviewer,

Thank you for the work done in reviewing our article. We tried to take into account all your comments without exception. Special thanks to you for the changes made to the list of references. For us, this significantly expanded the geography of knowledge.

In terms of fixes:

  • line 157 – remark «Commonwealth of Independent States should be written instead of CIS»;
  • line 196 – remark «the method of determining fatigue life should be slightly more detailed».

Separately, I thank you for the edit with the limit of fatigue. We also calculated them based on the experimental data of JSC "Ukrainian Energy Machines".

Reviewer 2 Report

The reviewed paper concerns with the thrust bearing unit of a high power hydrogenerator at the operational loading conditions. The FE code for the thrust bearings and their chambers is applied. The cause-and-effect relationships of emerging and development of defects are established. Existing methods for calculating the stressed state of a thrust bearing in the classical formulation for a stationary mode of operation are considered.

While the reviewed paper is interesting for a potential reader and worth of publishing, it needs only minor revision.

1. The phrase in Cyrillic letters in raw 15 should be either removed or reproduced in English.

2. Provide citation to sketches in Fig. 1.

3. Raw 71: "...shall operate.." --> "...should operate..."

4. Raw 74: "...life shall overcome..." --> "...life should overcome..."

5. Raw: 210: "...design shall occur.."  --> "...design should occur..." etc.

6. The authors should somewhere note  "It should also be noted that there are more refined theories for accounting material degradation due to fatigue than suggested by the adopted homogeneous elasticity, e.g. at cyclic loadings the surface degradation may result in necessity to consider materials with the functionally grading (FG) properties [9]" .   The corresponding citation can be found, for example, in DOI: 10.1007/s00033-019-1132-0

Author Response

Dear Reviewer,

Thank you for your work. All your comments have been taken into account by us.

In terms of fixes:

  • line 71 – remark "...shall operate.." --> "...should operate...";
  • line 74 – remark "...life shall overcome..." --> "...life should overcome...";
  • line 74 – remark "...design shall occur.." --> "...design should occur...".